# Exciton Superposition across Moiré States in a Semiconducting Moiré Superlattice

Zhen Lian[1,8], Dongxue Chen [1,8], Yuze Meng[1], Xiaotong Chen[1], Ying Su[2], Rounak Banerjee[3], Takashi Taniguchi [4], Kenji Watanabe [5], Sefaattin Tongay [3], Chuanwei Zhang [2], Yong-Tao Cui [6] ✉ & Su-Fei Shi [1,7] ✉

Moiré superlattices of semiconducting transition metal dichalcogenides enable unprecedented spatial control of electron wavefunctions, leading to emerging quantum states. The breaking of translational symmetry further introduces a new degree of freedom: high symmetry moiré sites of energy minima behaving as spatially separated quantum dots. We demonstrate the superposition between two moiré sites by constructing a trilayer WSe₂/ monolayer WS₂ moiré heterojunction. The two moiré sites in the first layer WSe₂ interfacing WS₂ allow the formation of two different interlayer excitons, with the hole residing in either moiré site of the first layer WSe₂ and the electron in the third layer WSe₂. An electric field can drive the hybridization of either of the interlayer excitons with the intralayer excitons in the third WSe₂ layer, realizing the continuous tuning of interlayer exciton hopping between two moiré sites and a superposition of the two interlayer excitons, distinctively different from the natural trilayer WSe₂.

Design and control of symmetries can lead to symmetry-protected states that are promising to revolutionize the field of quantum materials. For example, breaking the inversion symmetry or time-reversal symmetry can lead to Weyl semimetals[1]. The inversion symmetry breaking in transition metal dichalcogenides (TMDCs) gives rise to a valley degree of freedom that is promising for valleytronics and quantum information science based on valley-spin[2], which can be accessed through chiral light.

The recent emergence of semiconducting TMDC superlattices[3–9], which are constructed through twisted TMDCs with a lattice mismatch or twist angle, enables spatial control of the excitons in two-dimension (2D) with the tunable periodicity of 1–10 nm and ushers in unprecedented opportunities in engineering electrons and excitons, leading to intriguing correlated electronic states[3,6,10–15], the array of quantum emitters, and correlated exciton states resulting from flat excitonic band[16–18].

Translation symmetry breaking in TMDC moiré superlattices introduces a new degree of freedom: moiré sites, the high symmetry points in a moiré supercell, which can be local energy minima and act as quantum dots[5,19,20] that can confine electrons and excitons, as schematically shown in Fig. 1c. In addition, these high symmetry points are protected by the three-fold rotation symmetry and possess the unique valley degree of freedom through the pseudo angular momentum conservation[4,19]. As a result, coupling and hybridization of these high symmetry points will usher in new venues toward quantum information storage and processing. However, unlike the energy degeneracy of different valleys (K and K′), the energy barrier between different moiré sites (on the order of 10 s' meV), along with their spatial separation, greatly suppresses the direct coupling between these moiré sites.

Here, we demonstrate the superposition between two different moiré sites by introducing a layer degree of freedom to the TMDC

---

[1]Department of Chemical and Biological Engineering, Rensselaer Polytechnic Institute, Troy, NY 12180, USA. [2]Department of Physics, University of Texas, Dallas, TX 75083, USA. [3]School for Engineering of Matter, Transport and Energy, Arizona State University, Tempe, AZ 85287, USA. [4]International Center for Materials Nanoarchitectonics, National Institute for Materials Science, 1-1 Namiki, Tsukuba 305-0044, Japan. [5]Research Center for Functional Materials, National Institute for Materials Science, 1-1 Namiki, Tsukuba 305-0044, Japan. [6]Department of Physics and Astronomy, University of California, Riverside, California 92521, USA. [7]Department of Electrical, Computer & Systems Engineering, Rensselaer Polytechnic Institute, Troy, NY 12180, USA. [8]These authors contributed equally: Zhen Lian, Dongxue Chen. ✉e-mail: yongtao.cui@ucr.edu; shis2@rpi.edu

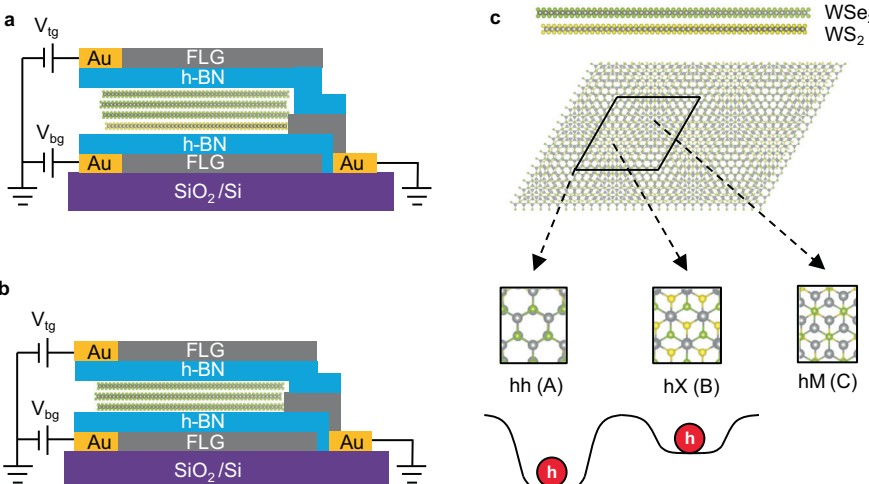

**Fig. 1 | Moiré site degree of freedom. a** and **b** are schematics of 3 L WSe$_2$/ 1 L WS$_2$ moiré heterojunction device and natural trilayer WSe$_2$ devices, respectively. Both devices are in a dual-gate configuration. **c** is the schematic of the WSe$_2$/WS$_2$ moiré superlattice, with three high symmetry points of C3 symmetry shown as hh, hX and hM. The naming convention of hh, hX and hM corresponds to aligning the hexagon center of the hole layer (WSe$_2$) with the hexagon center (h), the chalcogen atom (X), and the metal atom (M) of the electron layer (WS$_2$)[20], which we also call as A, B and C moiré sites for convenience. A and B sites are energy minima for holes and behave as quantum dots that confine carriers and excitons. We use the holes for illustration in **c**, but the trapping of electrons and excitons will be similar.

moiré superlattice. It is well known that the two neighboring layers of 2-H TMDC flakes, due to the intralayer inversion symmetry breaking, possess a layer degree of freedom that acts as pseudospins alternating in odd and even layers[2,21]. In an angle-aligned trilayer WSe$_2$/monolayer WS$_2$ heterojunction (3L WSe$_2$/1L WS$_2$), new types of interlayer excitons emerge, with holes residing in the first WSe$_2$ layer either trapped in moiré A or B site (Fig. 1c)[5], and electrons with the same pseudospin residing in the third WSe$_2$ layer. In particular, we find that these two interlayer excitons can hybridize through coupling with the intralayer excitons in the third WSe$_2$ layer. The resulting hybridized exciton inherits both the large oscillator strength from the intralayer excitons and the sensitive electric field dependence from the moiré interlayer excitons[7,22–24]. More interestingly, by applying an electric field, we can drive the transition between the two interlayer moiré excitons' hybridization with the intralayer exciton in the third WSe$_2$ layer, enabling the continuous tuning of hopping of the interlayer exciton from 100% at one moiré site to 100% at the other, which is otherwise suppressed. In between the transition points, we obtain an excitonic complex that is the superimposition of the interlayer excitons that are otherwise localized at moiré A and B sites.

## Results and Discussions
### Gate dependence of reflectance contrast
The schematic of the 3L WSe$_2$/1L WS$_2$ moiré heterojunction is shown in Fig. 1a, which is fabricated into a dual-gated device structure in which the doping and electric field can be independently controlled. We also fabricated a device of a dual-gated 2-H phase trilayer WSe$_2$ (3L WSe$_2$) (schematically shown in Fig. 1b) for the control study.

The doping-dependent optical reflectance contrast spectra of the 3L WSe$_2$/1L WS$_2$ heterojunction device are shown in Fig. 2e, which is evidently different from that of the natural trilayer (3L) WSe$_2$ device (Fig. 2f). The most pronounced resonance for the natural trilayer WSe$_2$ (Fig. 2f) is the intralayer exciton resonance X$_A$, which is at ~ 1.70 eV at zero doping, redshifted compared to the A exciton resonance in monolayer WSe$_2$ (~1.73 eV)[25]. X$_A$ is redshifted linearly for both n and p doping in a symmetric fashion, with a slope of ~1.3 meV/10$^{12}$ cm$^{-2}$. IX$_{3L}$ are the interlayer excitons with the hole and electron separated in the first and third WSe$_2$ layer, which have two degenerate modes as schematically shown in Fig. 2c and are named as IX$_{3L}^+$ and IX$_{3L}^-$ ("+" and "−" denote the direction of the dipole moment in the sample coordinate. The direction of the positive electric field or dipole moment is

defined as from the top gate to the back gate in 3 L WSe$_2$, and from WSe$_2$ to WS$_2$ in 3L WSe$_2$/ 1 L WS$_2$). IX$_{3L}^{2s}$ is the 2s state of the IX$_{3L}$. The natures of IX$_{3L}$ and IX$_{3L}^{2s}$ become obvious in our later discussion of the electric field dependent reflectance contrast spectra. Zoom-in of Fig. 2f with enhanced contrast is plotted in Fig. S2 to show IX$_{3L}$ and IX$_{3L}^{2s}$ more clearly. Accompanying X$_A$ is a less pronounced resonance X$_A'$ with a larger slope (2.7 meV/10$^{12}$ cm$^{-2}$). X$_A'$ is likely the exciton resonance of the middle (second) layer WSe$_2$ and is not the focus of this work (see detailed discussion in Supplementary Section 14).

In the optical reflectance contrast spectra of 3L WSe$_2$/1L WS$_2$ moiré heterojunction (Fig. 2e), there is an exciton resonance (X$_M^I$) located at the lower energy side of X$_A$ (~1.667 eV), which is only observable in angle-aligned 3L WSe$_2$/1L WS$_2$ heterojunctions but absent in heterojunctions with large twist angles (See supplementary section 10 for detailed discussion). X$_M^I$ is the previously discovered moiré intralayer exciton peak in the first layer WSe$_2$ interfacing WS$_2$, with the exciton trapped at the moiré A site. The doping dependence of X$_M^I$ clearly show the signature of the correlated insulating states at the filling factor of 1 and −1, corresponding to one electron and one hole per moiré superlattice, which was discussed in our previous publication[26]. On the p-doping side, the exciton resonances of X$_A$ and X$_A'$ are labeled as such due to their similar behaviors compared with that from the trilayer WSe$_2$ (Fig. 2f), with a redshift slope of 1.0 and 2.1 meV/10$^{12}$ cm$^{-2}$, respectively. The n-doping side is different because the electrostatically introduced electrons are in the WS$_2$ layer instead of the WSe$_2$ layers due to the type II alignment, leaving the WSe$_2$ layers charge-neutral. We identify the X$_A$ and X$_A'$ in the n-doping side through their slopes as well, 1.0 and 2.1 meV/10$^{12}$cm$^{-2}$, respectively, the same as those in the p-doping region. The abrupt blueshift of the X$_A$ in the n-doping side (starts at $n > 1$ and resonant energy around 1.725 eV) is likely due to the built-in electric field on WSe$_2$ layers arising from the electron accumulation in WS$_2$. We leave the related discussion in Supplementary Information Section 15. The focus of our work here is on the interlayer excitons within the 3L WSe$_2$ of the 3L WSe$_2$/1L WS$_2$ moiré heterojunction, with their schematics shown in Fig. 2a, b. The IX$_{3L}^+$ branch is visible and pronounced at the blue arrow in Fig. 2e, partially because it hybridizes with intralayer excitons and gains some oscillator strength but also because it retains the extended nature of interlayer excitons, hence sensing dielectric environment change associated with the Mott insulator transition at filling of one electron per moiré superlattice

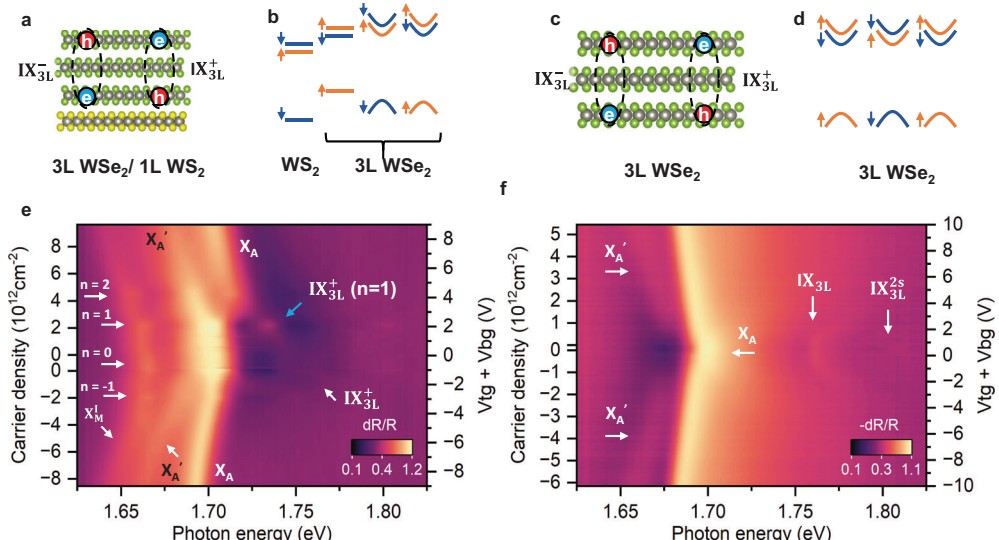

**Fig. 2 | Doping-dependent reflectance contrast spectra of 3L WSe₂/ 1L WS₂ and natural trilayer WSe₂ Devices. a**, **b** and **e** are the schematic atomic structure, band alignment, and the doping-dependent reflectance contrast spectra of 3L WSe₂/ 1L WS₂ measured from device D1. **c**, **d** and **f** are the schematic atomic structure, band alignment, and the doping-dependent reflectance contrast spectra of natural trilayer WSe₂ measured from device D2. The blue arrow in **e** denotes the enhanced reflection signal of the hybridized exciton (interlayer exciton $IX_{3L}^+$ hybridized with intralayer exciton) at the Mott insulator state at $n = 1$.

($n = 1$). The nature of both resonances are revealed in our later discussion of the electric field dependence study.

### Interlayer hybridized excitons in 3L WSe₂

The electric field-dependent reflectance contrast spectra of the trilayer WSe₂ device is shown in Fig. 3a, which is symmetric about the electric field due to its symmetric structure. The most noticeable feature is the "cross" pattern originating from the electric field evolution from interlayer exciton $IX_{3L}$. The slope of each branch of the cross is roughly the same. These are arising from the Stark shift of the interlayer exciton $IX_{3L}$, with the two degenerate modes ($IX_{3L}^-$ and $IX_{3L}^+$) shifting oppositely under an electric field due to the dipole moment of opposite polarity. The Stark energy shift can be expressed as $\Delta E = -edF$, where F is the local electric field, e is the electron charge, and d is the electron and hole separation. We extract the value of d to be about 1.26 nm for both $IX_{3L}^-$ and $IX_{3L}^+$, which is about twice that of interlayer exciton dipole moment in WSe₂/WS₂ (0.7 nm)[7], confirming that the electron and hole of interlayer exciton occupy the two outside WSe₂ layers in a natural trilayer WSe₂.

The level avoiding at the intralayer exciton A (-1.70 eV) in Fig. 3a arises from the hybridization of the interlayer exciton and intralayer exciton. In the 2H trilayer WSe₂, there is significant tunneling of holes between the first and third layer WSe₂ as they have the same valley-layer pseudo spin, allowing the hybridization of the interlayer excitons with the intralayer excitons in either the first or third WSe₂ layer[27], as schematically shown in Fig. 3d. This hybridization can be well captured by a coupled two-level system, which is given by the following Hamiltonian in the basis of intralayer exciton and interlayer exciton:

$$\begin{bmatrix} X_a & \Delta \\ \Delta & X_i(F) \end{bmatrix} \quad (1)$$

where $X_a$ is the energy of the intralayer exciton, $X_i(F)$ is the energy of the interlayer exciton at a given electric field F, $\Delta$ is the coupling strength (see Supplementary Information Section 13 for details).

Take the positive electric field (direction defined in Fig. 3d) scenario as an example (Fig. 3c): a linearly dispersed interlayer exciton $IX_{3L}^+$ (white dotted line in Fig. 3c) and a non-dispersed intralayer exciton $X_A$ (black dotted line) can be used to well fit the observed hybridized spectra (red and blue dashed lines). From the fitting, we extract

the coupling strength to be 10.7 ± 0.3 meV, larger than the linewidth of the hybridized exciton (~ 9.0 ± 0.3 meV). The scenario of the negative electric field is similar, where the other interlayer exciton mode, $IX_{3L}^-$, hybridizes with the intralayer exciton ($X_A$) when the energy of the two excitons is tuned to resonance via the electric field. It is worth noting that we ignore the conduction band hybridization of the first and third layer WSe₂, which is theoretically predicted to be nonzero but orders of magnitude smaller than the holes[27]. The neglection of the conduction band hybridization is also justified by the electric-field-dependent reflectance contrast spectra of 3 L WSe₂/ 1 L WS₂, which is asymmetric about positive and negative electric fields (later discussion of Fig. 4).

The additional level avoiding at the energy around 1.79 eV in Fig. 3a is due to the hybridization of the interlayer exciton ($IX_{3L}$) with the 2s state of intralayer A exciton (Fig. 3a and Fig. S4a, b). The second level avoiding at higher energy (-1.80 eV) is due to the hybridization of the excited state of the interlayer exciton ($IX_{3L}^{2s}$) and 2s of the A exciton, which we enhance the contrast and show in Fig. S4a, b. It is interesting to note that the energy difference between the ground state and 2s of interlayer exciton $IX_{3L}$ is about 51 meV, smaller but at the same order of magnitude compared with the energy difference between 2s and 1s of A exciton for trilayer WSe₂ (-95 meV, Fig. S4a, b), suggesting the strongly bound nature of the interlayer exciton $IX_{3L}$. All these hybridization features are absent in a dual-gated nature bilayer WSe₂ (Fig. S8), which is AB stacked with two layers of different layer pseudospin, further confirming our interpretation. The electric-field-dependent reflectance contrast spectra of a 4 L WSe₂ device (Fig. S7) show similar hybridization features but with two "crosses" slightly shifted in energy, about 10 meV. According to the interpretation of the 3 L WSe₂ data, these two crosses are the two types of interlayer excitons from the 1st and 3rd layer WSe₂ and the 2nd and 4th layer WSe₂, which slightly shift in energy due to possible dielectric environment differences[28,29].

### Hybridized Excitons across Moiré States in 3 L WSe₂/ 1 L WS₂

We now turn to the study of the electric field-dependent reflectance contrast spectra of the 3 L WSe₂/ 1 L WS₂ moiré heterojunction, shown in Fig. 4c. The negative electric field side has some similarity compared with that from trilayer WSe₂, while the positive electric field side is significantly different. More specifically, the hybridized spectrum on the positive electric field side involves three exciton branches: two

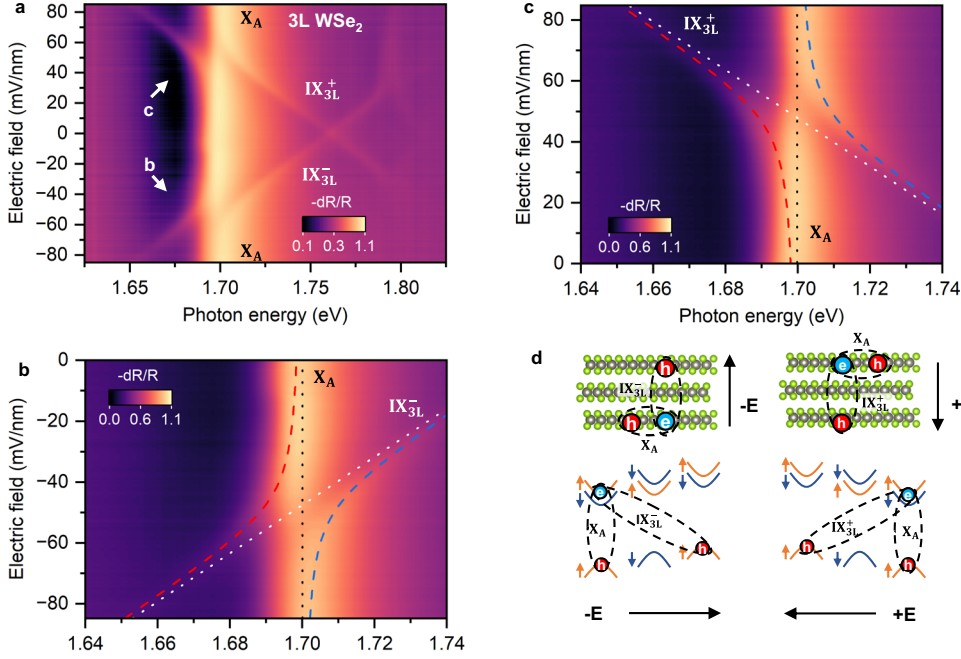

**Fig. 3 | Interlayer and intralayer exciton hybridization in natural trilayer WSe₂.** **a** shows the electric field dependence of reflectance contrast spectra measured from natural trilayer WSe₂ device (D2) plotted in log scale. **b** is the zoom-in of **a** in the region between 1.64 eV and 1.74 eV at negative electric fields plotted in linear scale. **c** is the zoom-in of **a** in the region between 1.64 eV and 1.74 eV at positive electric fields plotted in linear scale. The dashed red and blue lines show the fitting

result of the hybridized excitonic states obtained by fitting the peak positions with a two-level hybridization model. The white and black dotted lines are the energies of unhybridized intralayer and interlayer excitons obtained from the fitting. **d** shows the schematics of the interlayer and intralayer excitons involved in the hybridization, both in real space and the band alignment configurations.

dispersive (white dotted lines in Fig. 4e) and one non-dispersive (black dotted line in Fig. 4e) branch. The necessity of involving three exciton branches is also obvious from the derivative of Fig. 4e with respect to the electric field, as shown in Fig. S3d. The two dispersive excitons have a similar slope for the Stark shift, translating to electron and hole separations of $1.313 \pm 0.004$ nm and $1.609 \pm 0.004$ nm (fitting details in Supplementary Information Section 13). Therefore, they are the interlayer excitons, similar to $IX_{3L}^+$ in the natural trilayer WSe₂, with the hole in the first WSe₂ layer interfacing WS₂ and the electron in the third WSe₂ layer away from the interface. The two different interlayer excitons stem from the moiré coupling modified valence band of the first WSe₂ layer. As schematically shown in Fig. 4a and b, the moiré modulation folds the valence band of the first WSe₂ layer into moiré minibands. The two interlayer excitons correspond to holes occupying the two moiré minibands located at different moiré sites[20], which effectively behave as two spatially separated quantum dots. Each of them is located at an energy minimum at a high symmetry point within the moiré unit cell, which we call moiré A and B sites, respectively. We thus label these two interlayer excitons as $IX_{3L}^{+(A)}$ and $IX_{3L}^{+(B)}$. Since the WS₂ and first WSe₂ layer are aligned at 60 degrees (H stacked) as determined by the second harmonic generation (SHG) spectra (Fig. S5), the moiré A and B sites correspond to the $H_h^h$ and $H_h^X$ stacking configurations shown in Fig. 4b. The energy separation of the two interlayer excitons at zero electric field, 69 meV, represents the energy difference between the top two moiré minibands, if we ignore the difference in exciton binding energy. This value is consistent with the energy difference between the intralayer excitons trapped at moiré A and B sites in the WSe₂/WS₂ moiré superlattice, ~53 meV[26]. The remaining non-dispersive branch corresponds to the intralayer exciton, $X_A$, with both hole and electron in the third WSe₂ layer. Therefore, $IX_{3L}^{+(A)}$, $IX_{3L}^{+(B)}$, and $X_A$ hybridize by sharing the electron in the third WSe₂ layer. The above picture of hybridization involving two moiré interlayer excitons are confirmed by a control device (D4) of 3 L WSe₂/

1 L WS₂ in the dual-gate configuration, with an intentionally misaligned angle (20-degree) between WSe₂ and WS₂ layers. The electric-field-dependent reflectance contrast spectra (Fig. S11) indeed become symmetric about the electric field and similar to that of natural 3 L WSe₂, and they show no signs of interlayer moiré excitons ($IX_{3L}^{+(A)}$ and $IX_{3L}^{+(B)}$).

It is worth noting that direct tunneling between moiré A and B sites is suppressed due to the energy barrier, their spatial separation, and different stacking symmetry. Therefore, a direct hybridization between these two sites is difficult to achieve, unlike the degenerate valley-spin bands in TMDCs. However, with the assistance from the mobile intralayer exciton in the third WSe₂ layer, hybridization of moiré A and B sites is realized, and we can controllably tune the interlayer excitons $IX_{3L}^+$ between moiré A and B sites. In fact, the hybridized exciton notated with the cyan dashed line is a mixture of interlayer excitons localized at the moiré A site and B site, with the probability tunable from 100% at A to 100% at B site by controlling the electric field (Fig. 4f).

On the negative electric field side, the interlayer exciton involved in the hybridization is $IX_{3L}^-$, with the hole in the third layer WSe₂ not experiencing the moiré modulation. Meanwhile, the intralayer exciton in the 1ˢᵗ WSe₂ layer is modified by the moiré potential to have a lower energy of ~ 1.667 eV and is trapped at the moiré A site, which is labeled as $X_M^I$. As a result, hybridization occurs between $IX_{3L}^-$ and $X_M^I$. Their coupling strength is extracted to be $11.4 \pm 0.1$ meV. The interlayer exciton $IX_{3L}^-$ can also couple to the other moiré excitons from the 1ˢᵗ layer WSe₂, which contributes to the weak features in Fig. 4 and are shown with enhanced contrast in Fig. S4c, d.

The asymmetry of Fig. 4c between the n- and p-doping sides further justifies our neglection of conduction band hybridization: $IX_{3L}^-$ near hybridization region d in Fig. 4c goes directly through $X_A$, and $IX_{3L}^-$ near region e goes directly through $X_M^I$, with neither showing level avoiding. If the conduction band hybridization is significant, we should

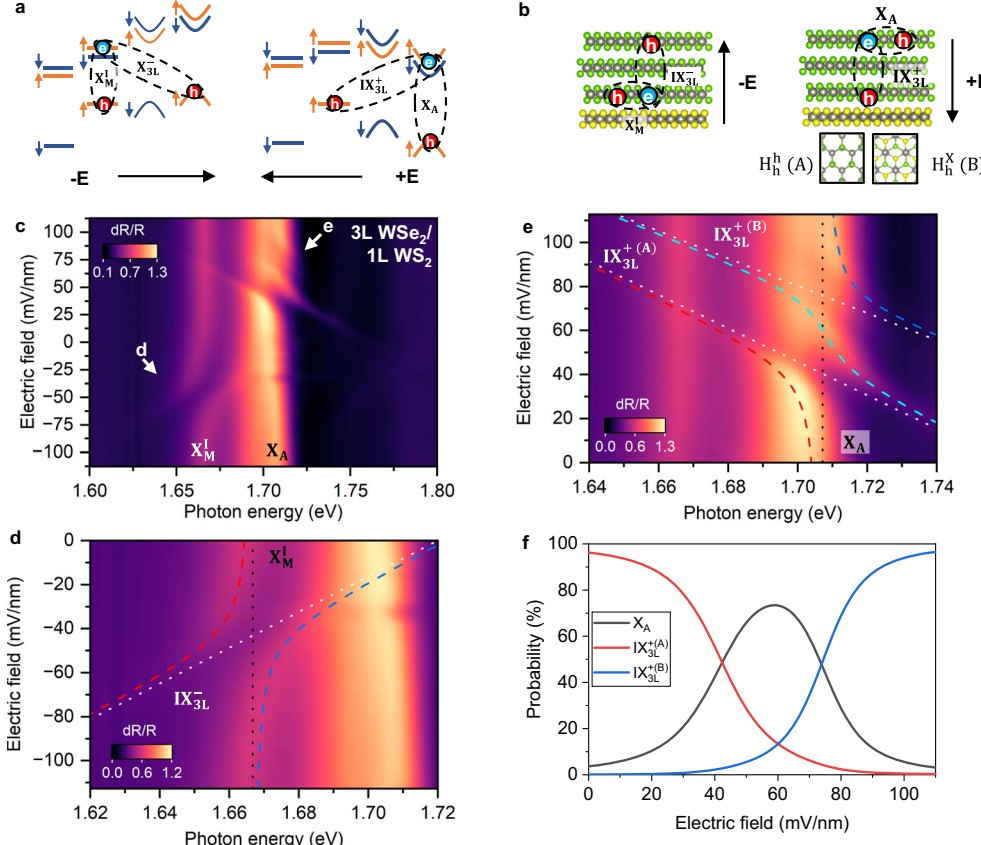

**Fig. 4 | Interlayer and intralayer exciton hybridization in the angle-aligned 3L WSe$_2$/1L WS$_2$ heterostructure. a** and **b** show the schematic band alignment and the real-space distribution of the hybridized excitons in 3L WSe$_2$/1L WS$_2$ moiré heterojunction. **c** shows the electric field dependence of reflectance contrast spectra measured from 3 L WSe$_2$/1L WS$_2$ device (D1). **d** is the zoom-in of **a** in the region between 1.62 eV and 1.72 eV at negative electric fields. **e** is the zoom-in of **a** in the region between 1.64 eV and 1.74 eV at positive electric fields. The dashed lines show the hybridized excitonic states obtained by fitting the peak positions. The dotted lines are the energies of unhybridized intralayer and interlayer excitons obtained from the fitting. A two-level hybridization model using one intralayer exciton and one interlayer exciton as bases is used to fit the peak positions in **d**, while a three-level hybridization model with one intralayer exciton and two different interlayer excitons is used to fit the peak positions in **e**. **f** shows the fractional composition of the hybridized exciton corresponding to the cyan line as a function of the electric field, expressed as the probability of each interlayer or intralayer exciton.

observe the hybridization of two interlayer excitons (due to moiré modulated conduction bands) and X$_A$. Similarly, the interlayer exciton from region e (IX$_{3L}^+$) in Fig. 4c should hybridize with X$_M^I$. We include a detailed discussion in Supplementary Information Section 16.

In summary, we have demonstrated a strategy to realize continuous tuning of interlayer exciton hopping between different moiré sites in 3 L WSe$_2$/ 1 L WS$_2$ moiré superlattices. These additional degrees of freedom enable the formation of a tunable honeycomb lattice of excitons with exciting opportunities for engineering new quantum states. For example, considering the large spin-orbit coupling in TMDCs, the continuous tuning of the hopping can be potentially exploited for constructing Dirac and Weyl modes of excitons, as well as the topologically protected edge states connecting these modes[19]. Our demonstration of the superposition of excitons across the different moiré sites also inspires new venues of quantum information processing and harnessing the new moiré site degree of freedom for twistronics.

## Methods
### Sample Fabrication
We used the same dry pick-up method[30] as reported in our earlier work to fabricate TMDC heterostructures[17,26]. The gold electrodes are pre-patterned on the Si/SiO$_2$ substrate. The monolayer TMDC flakes, BN flakes, and few-layer graphene (FLG) flakes are exfoliated on silicon chips with 285 nm thermal oxide. The thickness of BN flakes was determined by atomic force microscopy (AFM). The layer numbers of WSe$_2$ flakes were identified by optical contrast with the assistance of second-harmonic generation (SHG). Top BN and bottom flakes with equal thickness were intentionally used for devices D1, D2, and D3. The polycarbonate (PC)/ polydimethylsiloxane (PDMS) stamp was used to pick up TMDC monolayer and other flakes sequentially. The alignment of each layer is achieved under a home-built microscope transfer stage with the rotation controlled with an accuracy of 0.02 degrees. The PC is then removed in the chloroform/isopropanol sequence and dried with nitrogen gas. The final constructed devices were annealed in a vacuum (<10$^{-6}$ torr) at 250 °C for 8 hours.

### Optical Measurements
During the optical measurements, a home-built confocal imaging system was used to focus the laser onto the sample (with a beam spot diameter ~ 2 μm) and collect the optical signal into a spectrometer (Princeton Instruments). The reflectance contrast measurement was performed using a supercontinuum laser source (YSL photonics). A relative flat reflectance background $R_0$ was obtained by fitting the reflectance spectrum at high hole-doping level with a polynomial function for each measured spot (see Supplementary Section 11 for details). The reflectance contrast is defined as $\frac{dR}{R} = \frac{R - R_0}{R_0}$. The reflectance contrast from device D1 and D2 are added by 0.3 and −0.3 for better presentation in the log scale. All optical spectroscopy measurements were performed at the temperature of 10 K with a Montana cryostat. The polarized SHG measurements were performed with a

pulsed laser excitation centered at 900 nm (Ti: Sapphire; Coherent Chameleon) with a repetition rate of 80 MHz and a power of 80 mW. The crystal axes of the sample were fixed. A half-waveplate was placed between the beam splitter and the objective and was rotated to change the polarization angles of both the excitation laser and the SHG signal.

## Doping and Electric Field Calculations

The density of carriers introduced by the electrostatic gating is given by $n_e(n_p) = C_{tg}(V_{tg} - V_{tg}^0) + C_{bg}(V_{bg} - V_{bg}^0)$, where $C_{tg}(C_{bg})$ are the geometry capacitance of the top gate (back gate) and $V_{tg}(V_{bg})$ are the top gate (back gate) voltage. $V_{tg}^0$ and $V_{bg}^0$ are the onset gate voltages of the top gate and the back gate, determined experimentally from the regions where the 2s peaks remain visible. The electrical field in the TMDC is given by $F = \varepsilon_{BN}/\varepsilon_{TMDC}(V_{tg}/2d_1 - V_{bg}/2d_2)$, where $d_1(d_2)$ is the thickness of the top (bottom) BN determined by atomic force microscopy, $\varepsilon_{BN} = 3.5$ and $\varepsilon_{TMDC} = 7.2$ are the relative dielectric constants of h-BN and TMDC, respectively[31,32].

## Data availability

The data in Figs. 1–4 are provided in the source data files. All other data that support the plots within this paper and other findings of this study are available from the corresponding author upon reasonable request. Source data are provided with this paper.

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

## Acknowledgements

Z.L. and S.-F.S. acknowledge support from NYSTAR through Focus Center-NY–RPI Contract C180117. The device fabrication was supported by the Micro and Nanofabrication Clean Room (MNCR) at Rensselaer Polytechnic Institute (RPI). S.-F. S., D.C., and X.C. also acknowledge the support from NSF Grant DMR-1945420, DMR-2104902, and ECCS- 2139692. Y.-T.C. acknowledge support from NSF under award DMR-2104805. The optical spectroscopy measurements were supported by a DURIP awards through Grant FA9550-20-1-0179 and FA9550-23-1- 0084. S.T. acknowledges DMR-2111812 for materials development, DMR-2206987 for structural characterization and magnetic impurity tests, DMR-2052527 for electronic tests, CMMI-2129412 for structure-performance-property relations, and DOE SC-SC0020653 for initial excitonic characterization. K.W. and T.T. acknowledge support from JSPS KAKENHI (Grant Numbers 19H05790, 20H00354, and 21H05233). Y.S. and C.Z. acknowledge support from NSF PHY-2110212, PHY-1806227, ARO (W911NF17-1-0128), and AFOSR (FA9550-20-1-0220).

## Author contributions

S.-F.S conceived the project. Z.L., D.C. and Y.M. fabricated devices. Z.L. performed measurements. M.B. and S.T. grew the TMDC crystals. T.T. and K.W. grew the BN crystals. S.-F. S., Y.-T.C., Z.L. and D.C. analyzed the data. S.-F. S. supervised the project. S.-F. S. and Y.-T. C. wrote the manuscript with input from all authors.

## Competing interests

The authors declare no competing interest.
