## [Peer Review File · Nature Communications]

Reviewers' Comments:

Reviewer #1:

Remarks to the Author:

The paper by Zhen Lian et al. presents an optical study of the WSe₂ trilayer where the authors observe the hybridization of interlayer excitons localized in moiré sites when the trilayer is placed on the WS₂ monolayer. The latter monolayer, when oriented at 60 degrees with respect to the trilayer, generates the well-known honeycomb structure (moiré sites) where intralayer excitons are localized in Moiré sites. A comparison with other structures (trilayer only or another angle orientation) where there is no moiré effect is proposed. I think that all the experimental details are well described and understood. The results are overall reliable and interesting. In my opinion, it gives sufficient results to meet the criteria of Nature Communication. The paper can be published, although I have some comments or questions.

The extended Fig 6 is certainly very helpful to believe in the two-level hybridization. To confirm this, please provide the same data treatment for the trilayer where only one level is observed.

There is evidence that only the valence band is affected by hybridization. Is this based only on experimental evidence or is there a theoretical study confirming this fact? In 3L/1L structure, there is a change in the conduction band and the valence band (mainly the valence band probably); is this related to the impact on hybridization? This may be interesting if a theoretical background (or references) is added.

An equivalence between moiré and quantum well is given. Would it be better to give an equivalence with quantum dots?

The interlayer excitons lifetime can be affected by the out of plane field. Could it impact the results ?

I understand that the authors want to be trendy and use many of the "thing-tronics" words. As physicists, I think we should limit ourselves to new fancy words. Here, the word "moiretronics" joins a long list of similar terms. I recommend that the authors and editor avoid this type of word or, in the worst case, replace "moiretronics" with "twistronics", which is probably similar and already used in the literature...

Please use the sample as a reference instead of laboratory coordinates which might be unclear.

Fig 4e ; no grey dotted line is visible. Two white dotted lines can be sufficient.

P6 : X^i ,down_3L near region e goes (not go).

Reviewer #2:

Remarks to the Author:

The authors Zhen Lian et al. realize in their paper "Exciton Superposition across Moiré States in a Semiconducting Moiré Superlattice" interlayer excitons in a dual-gated 3L WSe₂ /1L WS₂ heterostructure with the electron moving freely in the outer WSe₂ layer and the hole being trapped in either the A or B moiré site at the WSe₂ /WS₂ interface. By applying an external electric field they can tune the interlayer excitons into resonance with the intralayer exciton in WSe₂, effectively tuning the hybridization of inter- and intralayer excitons in the system. In comparison with natural trilayer WSe₂ without any moiré modulation the presented experimental study demonstrates the potential of using the moiré site degree of freedom for realizing and modulating new quantum states.

This opens up an exciting direction towards realizing tunable excitonic systems whose properties can be additionally engineered using yet another degree of freedom, the moiré sites. The experimental approach of combining interlayer excitons in trilayers with an intentionally superimposed moiré potential is original and I believe this paper provides an interesting direction to the field of interlayer exciton physics in 2D materials. However, to satisfy the publication

standards of Nature Communications, the manuscript requires some revision/restructuring and addressing a few small technical points:

1. The authors describe that the moiré sites “can be local energy minima and act as quantum wells that can confine electrons and excitons, as schematically shown in Fig. 1c”, however, the schematic illustration indicates the trapping of holes.
2. The authors claim that in an angle-aligned 3L WSe₂ /1L WS₂ new interlayer excitons form. However, interlayer excitons are already observable in the 3L WSe₂. These interlayer excitons can also be tuned in resonance with the intralayer excitons and hybridize. The authors should clearly separate the description of the case with and without the additional WS₂ layer and its effect of introducing a moiré potential.
3. The label 3L/1L WSe₂/WS₂ is a bit confusing. I would suggest the following: 3L WSe₂ /1L WS₂.
4. The authors compare the intralayer exciton energy of the A-exciton in the trilayer directly with the A-exciton in the monolayer (I believe from a different device). As the exciton energy strongly depends on the dielectric environment and varies from sample to sample, this seems to not be a fair comparison.
5. Fig. 2c only indicates $\chi_{i,\text{down}}$ and no $\chi_{i,\text{up}}$. Also, I believe that at zero electric field, these two should be degenerate and it would make more sense to label the peak more generally as χ_i ?
6. Why is the slope of the intralayer excitons resonance linear with n ? How does this compare with the case of a monolayer where the exciton energy blueshifts with increasing density, following the exciton-polaron picture?
7. The optical features in Fig. 2e,f are very difficult to identify – is there a clearer way of presenting the data, e.g. by using the derivative as suggested in the supplement? Also, a clear label of the blue arrow in Fig. 2e is missing. Can the authors elaborate a bit more why the $\chi_{i,\text{down}}$ introduces a new resonance by hybridizing with the intralayer exciton at a finite electron density, not present at zero n ?
8. The authors claim that the “hybridized exciton inherits both the large oscillator strength from the intralayer excitons and the sensitive electric field dependence from the moiré interlayer excitons”. Out of curiosity, are the authors able to extract the relative oscillator strengths of the individual resonances when tuning into resonance to support the claim?
9. The section describing the avoided crossing with the 2s state (“The additional level avoiding at the energy around 1.79 eV and ...”) is a very nice experimental observation but might distract from the main text. This could, in my opinion, be moved to the supplement.
10. Would it be possible to indicate the Zoom-in-ranges a bit more clearly in Fig. 3 and 4?
11. Are the extracted dipole moments fitted before any interaction? The interaction might change the actual intrinsic Stark shift value. Also, I believe it is sufficient to only state the value for the Stark shift in terms of nm, rather than providing two values describing the same slope.
12. There is a typo “valance band” in the same section.
13. How strong is the interaction with the intralayer exciton with electric field? If it is true that the interaction is only with one of the intralayer excitons should one be able to distinguish between intralayer excitons in layer 3 and layer 1,2? Are the authors able to resolve any splitting here?
14. Could the authors elaborate a bit more on the probability of tuning between the A and B site as this seems to be one of the main statements. How does this tuning between A and B manifest itself in detail, how is it calculated?
15. The sections “We want to point out that the neglect of conduction band hybridization...” and “We further fabricated a control device...” seem a little detached from the main text and could be moved to the supplement. They nicely support the claim of the main text but could be distracting at the end of the main text.
16. What do the authors mean by continuous tuning of interlayer exciton hopping between different moiré sites?

Reviewer #3:

Remarks to the Author:

The manuscript presents reflection measurements of a van der Waals heterostructure composed of trilayer WSe₂ on monolayer WS₂. The authors investigate the hybridization of the interlayer exciton in trilayer WSe₂ with Moiré excitons at the WS₂/WSe₂ interface, taking advantage of the strong hybridization between interlayer and intralayer excitons in homotrilayer WSe₂. The data show two hybridization branches in the WS₂/WSe₂ device, attributed to the two Moiré

exciton branches hybridizing with the intralayer exciton. The key claim is that the two interlayer excitons become hybridized mediated via intralayer excitons. However, the evidence for the hybridization between interlayer excitons is not strong. Therefore, I cannot recommend the manuscript for publication in its current form. Below are my specific comments:

The central conclusion of the work relies on Fig. 4. However, the data quality in Fig. 4 is not high enough to give strong confidence in the analysis. For instance, the intralayer exciton becomes broad, making it challenging to identify the peak positions of various excitons. The choice of the cyan line can be arbitrary, undermining the analysis in Fig. 4f. The choice of intralayer exciton positions also

Suppose we assume the analysis and the extracted values from the coupled oscillator model are correct. Still, no direct evidence indicates that the two interlayer excitons come from different Moiré sites. Control experiments using larger twist angles are suggestive but not conclusive. Any effects splitting the WSe₂ valence band at the interface can lead to similar observations.

Why is the intralayer Moiré exciton not observed if the interlayer Moiré exciton can be observed?

How uniform is the sample, and are there any spatial variations in their optical properties?

Minor comments:

Why does the 2s interlayer exciton have a different dipole moment than the 1s exciton?

The analogy to the BCS model seems strange. This can happen for any coupled state; and there is no pairing here.

We sincerely thank the reviewers for their time and efforts, and we greatly appreciate the reviewers' recognition of our work and their constructive comments. Below we provide a point-to-point reply to reviewers' comments, and we have significantly revised our manuscript accordingly. With the revisions, we believe that we have fully addressed the reviewers' questions, and our manuscript is ready for publication in Nature Communications. We thank all the reviewers for helping us improve our manuscript.

Reviewer #1 (Remarks to the Author):

The paper by Zhen Lian et al. presents an optical study of the WSe₂ trilayer where the authors observe the hybridization of interlayer excitons localized in moiré sites when the trilayer is placed on the WS₂ monolayer. The latter monolayer, when oriented at 60 degrees with respect to the trilayer, generates the well-known honeycomb structure (moiré sites) where intralayer excitons are localized in Moiré sites. A comparison with other structures (trilayer only or another angle orientation) where there is no moiré effect is proposed. I think that all the experimental details are well described and understood. The results are overall reliable and interesting. In my opinion, it gives sufficient results to meet the criteria of Nature Communication. The paper can be published, although I have some comments or questions.

We thank the reviewer for the recognition of our work.

The extended Fig 6 is certainly very helpful to believe in the two-level hybridization. To confirm this, please provide the same data treatment for the trilayer where only one level is observed.

Here we provide the derivative of the reflectance contrast with respect to the electric field measured from device D2 in Fig. R1. We also include this figure in the revised manuscript as Extended Figure 3b.

Figure R1. Derivative of reflectance with respect to the electric field in natural trilayer (3L) WSe₂. The data for the derivative treatment is from Fig. 3b and 3c of the main text for (a) and (b), respectively.

There is evidence that only the valence band is affected by hybridization. Is this based only on experimental evidence or is there a theoretical study confirming this fact? In 3L/1L structure, there is a change in the conduction band and the valence band (mainly the valence band probably); is this related to the impact on hybridization? This may be interesting if a theoretical background (or references) is added.

We have experimental results supporting the much smaller effect of the conduction band hybridization, which is also supported by the previous theoretical work [PHYSICAL REVIEW RESEARCH 3, 043217 (2021)].

Experimentally, if the conduction band hybridization is involved, we expect another branch of hybridization (at the negative electric field regime but symmetric to the branch at the positive electric field) to show up in Fig. 3 and Extended Fig. 6. However, we did not observe such effects.

Theoretically, it is shown by the calculations in reference 27 [PHYSICAL REVIEW RESEARCH 3, 043217 (2021)] that in the R_h^h configuration (1st and 3rd WSe₂), the tunneling of holes is almost two orders of magnitude larger than that of electron tunneling at the K point.

TABLE VII. Tunneling strength as extrapolated from first-principle calculations for different homo- and heterobilayers.

		R_h^h	R_h^M	R_h^X	H_h^h	H_h^X	H_h^M
MoS ₂ -WS ₂	K _v (')	15.5 meV	0	0	51.1 meV	0	0
	K _c (')	7.5 meV	0	0	0	12.5 meV	0
	Λ _c	124.7 meV	175.6 meV	160.6 meV	206.7 meV	152.7 meV	119.8 meV
	Γ _v	215.9 meV	387.8 meV	392.0 meV	373.4 meV	344.6 meV	230.8 meV
MoSe ₂ -WSe ₂	K _v (')	19.4 meV	0	0	57.8 meV	0	0
	K _c (')	9.9 meV	0	0	0	19.1 meV	0
	Λ _c	139.6 meV	186.7 meV	172.6 meV	212.1 meV	165.1 meV	138.8 meV
	Γ _v	200.4 meV	365.2 meV	367.3 meV	349.0 meV	331.0 meV	219.0 meV
MoS ₂ -MoS ₂	K _v (')	15.4 meV	0	0	44.9 meV	0	0
	K _c (')	2.3 meV	0	0	0	6.1 meV	0
	Λ _c	131.5 meV	184.2 meV	184.2 meV	209.4 meV	169.6 meV	137.0 meV
	Γ _v	208.3 meV	391.0 meV	391.0 meV	352.9 meV	352.7 meV	242.5 meV
MoSe ₂ -MoSe ₂	K _v (')	19.3 meV	0	0	56.3 meV	0	0
	K _c (')	5.3 meV	0	0	0	11.9 meV	0
	Λ _c	147.1 meV	194.7 meV	194.7 meV	224.8 meV	177.8 meV	145.0 meV
	Γ _v	198.4 meV	365.8 meV	365.8 meV	357.4 meV	327.6 meV	208.3 meV
WS ₂ -WS ₂	K _v (')	21.1 meV	0	0	55.1 meV	0	0
	K _c (')	0.2 meV	0	0	0	0.8 meV	0
	Λ _c	152.7 meV	190.7 meV	190.7 meV	227.6 meV	180.6 meV	140.9 meV
	Γ _v	235.7 meV	362.2 meV	362.2 meV	359.5 meV	357.3 meV	232.1 meV
WSe ₂ -WSe ₂	K _v (')	23.1 meV	0	0	66.9 meV	0	0
	K _c (')	0.6 meV	0	0	0	1.5 meV	0
	Λ _c	155.2 meV	202.8 meV	202.8 meV	236.6 meV	184.0 meV	151.6 meV
	Γ _v	201.7 meV	356.0 meV	356.0 meV	346.9 meV	321.3 meV	208.9 meV

Figure R2. Theoretical calculated tunneling strength from PHYSICAL REVIEW RESEARCH 3, 043217 (2021).

An equivalence between moiré and quantum well is given. Would it be better to give an

equivalence with quantum dots?

The reviewer is correct in that the moiré site localization is more similar to the zero-dimensional quantum dot rather than the quantum well. We thank the reviewer for this valuable suggestion. We have revised our manuscript accordingly.

The interlayer excitons lifetime can be affected by the out of plane field. Could it impact the results ?

The lifetime of interlayer exciton will be affected by the out-of-plane field through the modulation of the electron-hole recombination rate. However, it will not impact our results.

The lifetime change is likely to show up in the photoluminescence (PL) spectra, as PL is a noncoherent process determined by absorption and emission, with the latter determined by the quantum yield and significantly impacted by the radiative and nonradiative lifetime.

However, the reflectance spectra in this work directly probe the coherent absorption process. As long as the transition is allowed, it will show up in the reflectance spectra.

I understand that the authors want to be trendy and use many of the “thing-tronics” words. As physicists, I think we should limit ourselves to new fancy words. Here, the word “moiretronics” joins a long list of similar terms. I recommend that the authors and editor avoid this type of word or, in the worst case, replace “moiretronics” with “twistronics”, which is probably similar and already used in the literature...

We thank the reviewer for the suggestion and have replaced the “moirétronics” with “twistronics”. We were trying to emphasize the utilization of the moiré-site degree of freedom while “twistronics” give rise to a broad range of exciting phenomena such as the flat band-induced correlated physics.

Please use the sample as a reference instead of laboratory coordinates which might be unclear.

We thank the reviewer for the insightful suggestion. We have redefined the coordinate using the sample as a reference. More specifically, we have defined the direction from WSe₂ pointing toward the WS₂ layer as the positive direction.

Fig 4e ; no grey dotted line is visible. Two white dotted lines can be sufficient.

We have changed Fig.4e as suggested. We thank the reviewer for helping us improve the quality of the figure.

Figure R3. Modified Fig.4e. The color of the grey dotted lines have been changed to white.

P6 : X^i ,down_3L near region e goes (not go).

We thank the reviewer for spotting the typo, which we have corrected. We have also proofread the manuscript again and corrected other possible typos.

Reviewer #2 (Remarks to the Author):

The authors Zhen Lian et al. realize in their paper “Exciton Superposition across Moiré States in a Semiconducting Moiré Superlattice” interlayer excitons in a dual-gated 3L WSe₂ /1L WS₂ heterostructure with the electron moving freely in the outer WSe₂ layer and the hole being trapped in either the A or B moiré site at the WSe₂ /WS₂ interface. By applying an external electric field they can tune the interlayer excitons into resonance with the intralayer exciton in WSe₂, effectively tuning the hybridization of inter- and intralayer excitons in the system. In comparison with natural trilayer WSe₂ without any moiré modulation the presented experimental study demonstrates the potential of using the moiré site degree of freedom for realizing and modulating new quantum states.

This opens up an exciting direction towards realizing tunable excitonic systems whose properties can be additionally engineered using yet another degree of freedom, the moiré sites. The experimental approach of combining interlayer excitons in trilayers with an intentionally superimposed moiré potential is original and I believe this paper provides an interesting direction to the field of interlayer exciton physics in 2D materials. However, to satisfy the publication standards of Nature Communications, the manuscript requires some revision/restructuring and addressing a few small technical points:

We greatly appreciate the reviewer’s recognition of our work. Below we reply to the comments in a point-to-point fashion.

1. The authors describe that the moiré sites “can be local energy minima and act as quantum wells that can confine electrons and excitons, as schematically shown in Fig. 1c”, however, the schematic illustration indicates the trapping of holes.

We use the electrons in the general sense and treat the holes as collective motions of electrons. As a result, we believe that the confinement works for both electrons and holes.

The reviewer is correct in noting that we only show holes in Fig.1c, as we mostly discuss the valence band hybridization in this work. To clarify that, we added the following statement in the caption of Fig. 1: “We use the holes for illustration, but the trapping of electrons and excitons will be similar.”

Also, reflecting on reviewer 1’s comments, it is more precise to change “quantum wells” to “quantum dots”. We thank both reviewers for their suggestions.

2. The authors claim that in an angle-aligned 3L WSe₂ /1L WS₂ new interlayer excitons form. However, interlayer excitons are already observable in the 3L WSe₂. These interlayer excitons can also be tuned in resonance with the intralayer excitons and hybridize. The authors should clearly separate the description of the case with and without the additional WS₂ layer and its effect of introducing a moiré potential.

The reviewer is correct in that the interlayer exciton exists in the natural trilayer WSe₂ (3L WSe₂). However, in the 3L WSe₂/1L WS₂ moiré heterojunction, due to the moiré potential modulation at the WSe₂/WS₂ interface [Nat Commun 13, 4810 (2022)], the electron or hole constituting the interlayer exciton will be localized at the high symmetry point (moiré A or B site as mentioned in the manuscript). This not only changes the interlayer exciton energy but also significantly changes

the exciton wavefunction distribution, leading to the new moiré interlayer exciton, in contrast to the delocalized interlayer excitons in 3L WSe₂.

We have revised the manuscript as following:

“In an angle-aligned trilayer WSe₂/monolayer WS₂ heterojunction (3L WSe₂/ 1L WS₂), new types of interlayer excitons emerge, with holes residing in the first WSe₂ layer and either trapped in moiré A or B site (Fig. 1c)³, and electrons with the same pseudospin residing in the third WSe₂ layer.”

3. The label 3L/1L WSe₂/WS₂ is a bit confusing. I would suggest the following: 3L WSe₂ /1L WS₂.

We thank the reviewer for the suggestion, and we have changed it to 3L WSe₂ /1L WS₂ throughout the revised manuscript.

4. The authors compare the intralayer exciton energy of the A-exciton in the trilayer directly with the A-exciton in the monolayer (I believe from a different device). As the exciton energy strongly depends on the dielectric environment and varies from sample to sample, this seems to not be a fair comparison.

We thank the reviewer for the question and take this opportunity to elaborate on our thoughts and reasoning.

We have indeed measured the reflectance contrast of monolayer WSe₂ and natural trilayer on the same device (D1, which contains regions of the monolayer WSe₂, natural trilayer WSe₂, and

Figure R4. Reflectance contrast spectrum of monolayer WSe₂ and natural trilayer WSe₂ measured from device D1.

the 3L WSe₂ /1L WS₂ moiré junction). The A-exciton resonance of monolayer WSe₂ and trilayer WSe₂ is 1.740 eV and 1.697 eV, respectively. We believe that the increased dielectric constant of the trilayer WSe₂ contributed to this redshift of the A exciton. This redshift has been observed consistently in other samples.

5. Fig. 2c only indicates Xi_{i,down} and no Xi_{i,up}. Also, I believe that at zero electric field, these two should be degenerate and it would make more sense to label the peak more generally as Xi?

The reviewer is probably referring to the doping-dependent spectra in Fig.2e. In 3L WSe₂/ 1L WS₂, the moiré coupling breaks the out-of-plane mirror symmetry of 3L WSe₂. As a result, the energies of Xi_{i,down} and Xi_{i,up} (renamed as IX_{3L}^+ and IX_{3L}^- in the revised manuscript) are no longer degenerate. The feature at around 1.76 eV corresponds to Xi_{i,down}, while Xi_{i,up} is at around 1.72 eV. This can be verified by the electric-field-dependent reflectance contrast spectra in Fig.4c. The two interlayer excitons with opposite dipole moments are clearly at different energies at zero electric field.

6. Why is the slope of the intralayer excitons resonance linear with n? How does this compare with the case of a monolayer where the exciton energy blueshifts with increasing density, following the exciton-polaron picture?

In our previous work [Nat. Phys. 18, 1171–1176 (2022)], we have estimated the blueshift of the repulsive exciton polaron in monolayer WSe₂ on device D1 (SI FigS6 c and d). Here we replot them as Figs. R5a,b. The repulsive exciton polaron roughly blueshifts linearly as a function of carrier density with a slope around 23 meV/10¹²cm⁻². As calculated in supplementary section 4, the slope of X_A' is around 2.7 meV/10¹²cm⁻² in 3L WSe₂ and 2.1 meV/10¹²cm⁻² in 3L WSe₂/ 1L WS₂, significantly smaller than that of the repulsion exciton polaron in monolayer WSe₂.

Figure R5. Blueshift of the repulsive exciton polaron in monolayer WSe₂ on device D1.

7. The optical features in Fig. 2e,f are very difficult to identify – is there a clearer way of presenting the data, e.g. by using the derivative as suggested in the supplement? Also, a clear label of the blue arrow in Fig. 2e is missing. Can the authors elaborate a bit more why the

$X_{i,\text{down}}$ introduces a new resonance by hybridizing with the intralayer exciton at a finite electron density, not present at zero n ?

The blue arrow in Fig. 2e is pointing at $X_{i,\text{down}}$ (renamed as IX_{3L}^+ in the revised manuscript) at the filling factor of $n = 1$. The $n = 1$ state in 3L $\text{WSe}_2/1\text{L } \text{WS}_2$ is a Mott insulator state according to our previous study [Nat Commun 13, 4810 (2022)]. We expect the dielectric environment to be changed significantly at $n = 1$ due to the Mott transition. The hybridized exciton, due to its extended wavefunction (inherited from interlayer exciton), is more sensitive to the local dielectric environment. Therefore, the resonance at $n = 1$ can be explained by the dielectric sensing of the hybridized exciton, similar to the 2s exciton sensing of correlated insulating states reported earlier [Nature 587, 214–218 (2020), Nat. Mater. 22, 175–179 (2023)]. We have explained this in the manuscript and also added a description in the caption of Fig. 1 of the main text.

The derivative of Fig. 2f is shown in Extended Fig. 3a. Here, we show the derivative of Fig. 2e in Fig. R6, which we included in the Extended Fig. 3c of the revised manuscript.

Figure R6. The derivative of reflectance contrast with respect to carrier density in device D1. The data for the derivative treatment is from Fig. 2e.

8. The authors claim that the “hybridized exciton inherits both the large oscillator strength from the intralayer excitons and the sensitive electric field dependence from the moiré interlayer excitons”. Out of curiosity, are the authors able to extract the relative oscillator strengths of the individual resonances when tuning into resonance to support the claim?

These statements are supported by previous experiments [Nat. Nanotechnol. 15, 901–907 (2020)][Nat. Nanotechnol. 16, 52–57 (2021)]. It was also shown that the interlayer excitons oscillator strength is typically two orders of magnitude smaller than intralayer excitons ([Science 376,406-410(2022)], [Nano Lett. 2017, 17, 2, 638–643]). Here, despite the complicated device structure, we estimate the oscillator strength difference between the intralayer exciton and hybridized exciton. The latter can be enhanced to roughly the same as that of the intralayer exciton, a two-order of magnitude increase from the expected value of interlayer excitons.

Here we provide an estimate of the relative oscillator strengths of the hybridized excitons based on peak intensities on the dR/R plot. We replot Fig.4c as Fig. R7a and show the linecut of dR/R at the electric field of 30 mV/nm and 55 mV/nm in Fig.R7 b and c, which is marked by the white dashed lines in a. In this region, the dR/R spectra can be fitted using 4 Lorentzian peaks (the fittings are also shown in Figs. R7b,c): intralayer moire exciton X_M^I , intralayer exciton from the middle WSe2 layer X_A^I , the low energy branch of the hybridized excitons, hX (LE), and the middle energy branch hX (ME).

The line cut in Fig.R7b is away from the center of the hybridization. hX (LE) possesses a large integrated intensity of 0.012 ± 0.001 , while hX (ME) shows a relatively small integrated intensity of $(8 \pm 1) \times 10^{-4}$. This observation is consistent with the expectation that hX (LE) has a large intralayer exciton component while hX (ME) mainly consists of the interlayer exciton.

As the electric field tunes the intralayer exciton and the interlayer into resonance in Fig.R7c, the hybridization is expected to be stronger. hX (LE) redshifts in Fig.R7c compared with Fig.R7b and shows a reduced peak intensity of $(8 \pm 1) \times 10^{-3}$, indicating the increased interlayer exciton component. Meanwhile, the integrated intensity of hX (ME) increases to $(9.3 \pm 0.7) \times 10^{-3}$ by hybridizing with the intralayer exciton. These results are consistent with the expected oscillator strength of interlayer excitons and intralayer excitons.

Figure R6. Integrated peak intensity analysis of hybridized excitons. (a) shows the same data in Fig.4c, with dashed lines indicating the positions of the line cuts. (b) and (c) show the four-Lorentzian-peak fitting of the dR/R spectra.

9. The section describing the avoided crossing with the 2s state (“The additional level avoiding at the energy around 1.79 eV and …”) is a very nice experimental observation but might distract from the main text. This could, in my opinion, be moved to the supplement.

We thank the reviewer for the suggestions. We have removed and rewritten these two paragraphs so that they are better fit into the flow the main text.

10. Would it be possible to indicate the Zoom-in-ranges a bit more clearly in Fig. 3 and 4?

We have added the following description in the caption of Fig.3 and 4:

“(b) is the zoom-in of (a) in the region between 1.64 eV and 1.74 eV at negative electric fields plotted in linear scale. (c) is the zoom-in of (a) in the region between 1.64 eV and 1.74 eV at positive electric fields, plotted in linear scale.”

“(d) is the zoom-in of (a) in the region between 1.62 eV and 1.72 eV at negative electric fields. (e) is the zoom-in of (a) in the region between 1.64 eV and 1.74 eV at positive electric fields”

11. Are the extracted dipole moments fitted before any interaction? The interaction might change the actual intrinsic Stark shift value. Also, I believe it is sufficient to only state the value for the Stark shift in terms of nm, rather than providing two values describing the same slope.

We had considered the interaction between the interlayer exciton and the intralayer exciton when we extracted the dipole moment. The details of the fitting can be found in supplementary section 3. Take the interaction between interlayer exciton and intralayer exciton in 3L WSe₂ as an example, and we use the A-exciton state and the interlayer exciton state as the bases of the two-level system. Based on the assumption that the energy of interlayer exciton shifts linearly under an out-of-plane electric field and the intralayer exciton exhibits zero Stark shift, the Hamiltonian of this two-level system can be expressed as:

$$H = \begin{bmatrix} Xa & \Delta \\ \Delta & Xi - e \cdot d \cdot F \end{bmatrix}$$

The two eigenvalues of H , denoted as $\xi_1(F)$ and $\xi_2(F)$, give the energies of the two hybridized excitons. The values of Xa , Δ , Xi and d are all set as fitting parameters and $\xi_i(F)$ is fitted with respect to the experimentally observed hybridized exciton energies to extract the fitting parameters. Therefore, the extracted d corresponds to the actual intrinsic dipole moment of the interlayer exciton in 3L WSe₂.

We have also fitted the slope of the hybridized excitons at large electric fields, which should be close to the dipole moment without coupling. The extracted interlayer distance is around 1.1 nm, close to the value extracted from the above model (1.26 nm).

Figure R8. The extracted interlayer distances from device D2 (3L WSe₂). The gray dots indicate peak positions extracted from the experimental data. The dashed lines indicate the slope of the hybridized excitons obtained by directly fitting the peak positions with a linear function of the electric field

12. There is a typo “valance band” in the same section.

We thank the reviewer for spotting the typo. We have corrected the spelling.

13. How strong is the interaction with the intralayer exciton with electric field? If it is true that the interaction is only with one of the intralayer excitons should one be able to distinguish between intralayer excitons in layer 3 and layer 1,2? Are the authors able to resolve any splitting here?

As mentioned in the main text and Supplementary Section 3, the coupling strength between intralayer exciton and interlayer exciton is around 10-11 meV for both 3L WSe₂ and 3L WSe₂/ 1L WS₂. In 3L WSe₂ the splitting of excitons in the first, second, and third layers are too small to be resolved. In 3L WSe₂/ 1L WS₂, the intralayer exciton energy in each layer is indeed different due to the interfacial moiré coupling and the difference of the dielectric environment. From the reflectance spectra of device D1(Fig. 4 of the main text), we determine the intralayer exciton energy to be 1.667 eV for moiré exciton X_M^I at the interfacial layer, ~ 1.70 eV for the A-exciton in the second layer, and ~ 1.71 eV for the A-exciton in the third layer.

14. Could the authors elaborate a bit more on the probability of tuning between the A and B site as this seems to be one of the main statements. How does this tuning between A and B manifest itself in detail, how is it calculated?

As discussed in Supplementary Section 3, we use the intralayer exciton (X_A), the interlayer exciton at A site ($X_i^{(A)}$) and the interlayer exciton at B site ($X_i^{(B)}$) as the bases of the three-level system in 3L WSe₂/ 1L WSe₂. The Hamiltonian can be written as:

$$H = \begin{bmatrix} Xa & \Delta 1 & \Delta 2 \\ \Delta 1 & Xi1 - e \cdot d1 \cdot F & 0 \\ \Delta 2 & 0 & Xi2 - e \cdot d2 \cdot F \end{bmatrix}$$

The three eigenfunctions of the Hamiltonian at a given electric field F is, therefore, a linear combination of the three bases:

$$\xi_j = \begin{bmatrix} c_{j1}(F) \\ c_{j2}(F) \\ c_{j3}(F) \end{bmatrix}$$

The values of $|c_{j1}(F)|^2$, $|c_{j2}(F)|^2$, $|c_{j3}(F)|^2$ give the composition of each hybridized exciton as the probability of each basis.

The eigenvalue E_j is fitted to the peak positions of the hybridized excitons as functions of electric field F to extract the model parameters (Xa , $Xi1$, $Xi2$, $\Delta 1$, $\Delta 2$, $d1$, $d2$). With the parameters known, we can then calculate the values of $|c_{j1}(F)|^2$, $|c_{j2}(F)|^2$, $|c_{j3}(F)|^2$ at a given electric field

F. In Fig.4f, we plotted the values of $|c_{21}(F)|^2$, $|c_{22}(F)|^2$, $|c_{23}(F)|^2$ (the composition of the mid-energy branch of the hybridized excitons).

15. The sections “We want to point out that the neglect of conduction band hybridization...” and “We further fabricated a control device...” seem a little detached from the main text and could be moved to the supplement. They nicely support the claim of the main text but could be distracting at the end of the main text.

We thank the reviewer for the suggestion and have moved the mentioned discussion to the revised SI.

16. What do the authors mean by continuous tuning of interlayer exciton hopping between different moiré sites?

We believe that this question is addressed in our reply to the reviewer’s question 14.

Reviewer #3 (Remarks to the Author):

The manuscript presents reflection measurements of a van der Waals heterostructure composed of trilayer WSe₂ on monolayer WS₂. The authors investigate the hybridization of the interlayer exciton in trilayer WSe₂ with Moiré excitons at the WS₂/WSe₂ interface, taking advantage of the strong hybridization between interlayer and intralayer excitons in homotrilayer WSe₂. The data show two hybridization branches in the WS₂/WSe₂ device, attributed to the two Moiré exciton branches hybridizing with the intralayer exciton. The key claim is that the two interlayer excitons become hybridized mediated via intralayer excitons. However, the evidence for the hybridization between interlayer excitons is not strong. Therefore, I cannot recommend the manuscript for publication in its current form. Below are my specific comments:

We thank the reviewer for reviewing our work, and we provide a point-to-point reply below.

The central conclusion of the work relies on Fig. 4. However, the data quality in Fig. 4 is not high enough to give strong confidence in the analysis. For instance, the intralayer exciton becomes broad, making it challenging to identify the peak positions of various excitons. The choice of the cyan line can be arbitrary, undermining the analysis in Fig. 4f. The choice of intralayer exciton positions also

The reviewers' comment seems to be unfinished here. We try to address the reviewer's questions in the following.

In our original Fig.4c, we have presented the data in log scale to manifest the subtle high-energy features on the spectra. However, this indeed blurs the main features of the spectra. Here we provide the color plot in Fig.4c plotted in linear scale, originally shown as extended Fig. 6a, as Fig. R9. The three-level hybridization can be clearly resolved in the plot. We have also replaced Fig. 4c with this version.

Figure R9. Fig. 4c plotted in linear scale.

We also emphasize that the choice of the interlayer exciton peak positions is not arbitrary. The peak positions can be extracted by searching the points with $d(dR/R)/dF = 0$. The changes in intralayer exciton energies are negligible under an out-of-plane electric field. Only the hybridized excitons will manifest themselves on the $d(dR/R)/dF$ plot. Fig.R10 shows the comparison of the extracted hybridized exciton peak positions and the $d(dR/R)/dF$ plot. The extracted peak positions in the full range of the reflectance spectra can be found in supplementary section 2 and are consistent with the peak positions on the color plot. Similar data treatment has been used in [Nature 595, 53–57 (2021)] to extract the features from the gate dependence of reflectance contrast spectra.

Figure R10. Extracted hybridized exciton peak positions overlaid on the $d(dR/R)/dF$ plot.

We hope we have convinced the reviewer that the quality of our data supports our claim.

Suppose we assume the analysis and the extracted values from the coupled oscillator model are correct. Still, no direct evidence indicates that the two interlayer excitons come from different Moiré sites. Control experiments using larger twist angles are suggestive but not conclusive. Any effects splitting the WSe_2 valence band at the interface can lead to similar observations.

This unique hybridization does not occur for trilayer WSe_2 and misaligned 3L WSe_2 /1L WS_2 heterojunction. We believe that is strong evidence that this hybridization is related to the moiré effect. Further analyses of Fig. 4 of the main text are also consistent with the moiré A and B site splitting.

Why is the intralayer Moiré exciton not observed if the interlayer Moiré exciton can be observed?

The intralayer moiré exciton is indeed observed. We have marked the positions of the first two intralayer moiré excitons (X_M^I and X_M^{II}) in Fig.R11. These features are not observed in the 20°-twisted sample shown in Extended Fig.9.

Figure R11. Intralayer moiré exciton in 3L WSe₂/ 1L WS₂ moiré superlattice. X_M^I and X_M^{II} label the positions of the first two intralayer moiré excitons. The data is presented in a log scale.

How uniform is the sample, and are there any spatial variations in their optical properties?

We have measured detailed field-dependent reflectance contrast spectra of another spot (spot B) from the 3L WSe₂/ 1L WS₂ region of device D1, which is similar to what is presented in the main text (Fig. R12). The major features discussed in the main text are also reproduced in data from another device (D3, Extended Figs 6a,b in the revised manuscript).

Figure R7. Electric field dependence of reflectance contrast spectra measured from point B of device D1. (a) shows the electric field dependence of reflectance contrast spectra from point B. (b) indicates the spots measured. The data in the main text is measured from point A.

Minor comments:

Why does the 2s interlayer exciton have a different dipole moment than the 1s exciton?

The 2s interlayer exciton is more extended than the 1s interlayer exciton, and we expect the dipole moment of the 2s interlayer exciton to be larger. However, the 2s interlayer exciton is not the focus of this work, and we will investigate it in the future work.

The analogy to the BCS model seems strange. This can happen for any coupled state; and there is no pairing here.

We thank the reviewer for this comment. We have removed the BCS model analogy from the main text.

Reviewers' Comments:

Reviewer #1:

Remarks to the Author:

The revised manuscript presents sufficient improvements to be published in Nature Communications. I am satisfied with the responses which have clarified the experimental observations.

Reviewer #2:

Remarks to the Author:

The authors have addressed my comments in the revised manuscript, provided a detailed response, and have improved their manuscript. Thus, I can now recommend this work for publication in Nature Communications.

Reviewer #3:

Remarks to the Author:

The authors have addressed my concerns regarding the validity and quality of the data well. Therefore, I am happy to recommend the publication of the paper in Nature Communications.

However, I believe it is crucial to include the identification of intralayer moire excitons in the paper (either in the main text or supplementary information). This identification would serve as compelling evidence for moire effects. In addition, for this set of data, it would be beneficial to present the measured spectrum instead of only showing gate-dependent colormap, which can sometimes be challenging to interpret.

REVIEWERS' COMMENTS

We thank all reviewers for their efforts again. We have addressed the remaining question in the reply below.

In addition, to comply with the format requirements of Nature Communications, we have moved the extended Figs. 1-10 in the original manuscript into the revised Supplementary Information as Figs. S1-9 and S11.

With these revisions, we believe that we have addressed all reviewers' questions, and our manuscript is ready for publication in Nature Communications. We thank all reviewers for helping us improve our manuscript.

Reviewer #1 (Remarks to the Author):

The revised manuscript presents sufficient improvements to be published in Nature Communications. I am satisfied with the responses which have clarified the experimental observations.

We thank the reviewer for the recognition of our work and the recommendation for publication.

Reviewer #2 (Remarks to the Author):

The authors have addressed my comments in the revised manuscript, provided a detailed response, and have improved their manuscript. Thus, I can now recommend this work for publication in Nature Communications.

We thank the reviewer for the recognition of our work and the recommendation for publication.

Reviewer #3 (Remarks to the Author):

The authors have addressed my concerns regarding the validity and quality of the data well. Therefore, I am happy to recommend the publication of the paper in Nature Communications.

However, I believe it is crucial to include the identification of intralayer moiré excitons in the paper (either in the main text or supplementary information). This identification would serve as compelling evidence for moiré effects. In addition, for this set of data, it would be beneficial to present the measured spectrum instead of only showing gate-dependent colormap, which can sometimes be challenging to interpret.

We thank the reviewer for the recognition of our work and the recommendation for publication.

We have included the discussion of intralayer moiré excitons in the revised supplementary section 10 and mentioned it in the main text. In addition, we have included the line plots of the

spectra in this section as Fig. S10.